# Long-Term Effects on Gonadal Function After Treatment of Colorectal Cancer: A Systematic Review and Meta-Analysis

**DOI:** 10.3390/cancers16234005

**Published:** 2024-11-29

**Authors:** Christiane Anthon, Angela Vidal, Hanna Recker, Eva Piccand, Janna Pape, Susanna Weidlinger, Marko Kornmann, Tanya Karrer, Michael von Wolff

**Affiliations:** 1OVA IVF, Clinic Zurich, 8005 Zurich, Switzerland; 2Division of Gynecological Endocrinology and Reproductive Medicine, University Women’s Hospital, Inselspital Bern, University of Bern, 3010 Bern, Switzerland; angela.vidal@insel.ch (A.V.); eva.piccand@bluewin.ch (E.P.); janna.pape@insel.ch (J.P.); susanna.weidlinger@insel.ch (S.W.); michael.vonwolff@insel.ch (M.v.W.); 3English Division, Wroclaw Medical University, Pasteura 1, 50-367 Wroclaw, Poland; hanna.recker@icloud.com; 4Department of General and Visceral Surgery, Surgery Center, Ulm University Medical Center, 89081 Ulm, Germany; marko.kornmann@uniklinik-ulm.de; 5Medical Library, University Library Bern, University of Bern, 3012 Bern, Switzerland; tanya.karrer@unibe.ch

**Keywords:** colorectal cancer, infertility, oncological treatment, FertiTOX, FertiPROTEKT

## Abstract

Colorectal cancer (CRC) is one of the most frequently diagnosed cancers, with more than 10% of cases occurring in young adults. Fertility is important for the Quality of Life. Counseling about fertility preservation measures is crucial before the start of gonadotoxic therapy. We, therefore, systematically analyzed the published literature on the gonadotoxic effects of CRC treatments in order to better counsel patients on the risk of infertility and the need for fertility preservation measures. The qualitative analysis included 22 out of 4420 studies. The meta-analysis included ten studies and showed an overall prevalence of clinically relevant gonadotoxicity of 23% (95% CI: 13–37%). In conclusion, this first meta-analysis evaluates the pooled prevalence of clinically relevant gonadotoxicity after CRC treatment. It provides clinically relevant information to counsel patients about the risk of infertility and the need to consider fertility preservation measures. The prevalence of gonadotoxicity was low in the case of chemotherapy only but rather high in the case of radiotherapy or radiochemotherapy. However, fertility preservation is also recommended in chemotherapy-only cases because dose-intensive follow-up treatments cannot be excluded and because extensive, longitudinal data on individual treatment effects are lacking. This review and meta-analysis give information about how cancer treatment can impact fertility and discuss available fertility preservation options.

## 1. Introduction

Colorectal cancer is one of the most frequently diagnosed cancers, accounting for about 10% of all newly diagnosed cases of cancer [1]. In many older studies, colon and rectal cancer are mainly described together as colorectal cancer. However, as colon and rectal cancer are different entities, they should be handled separately. The UICC 2003 (Union for International Cancer Control) defines rectal cancer as occurring less than 16 cm from the anocutaneous line [2,3]. Cancer that occurs more cranially is defined as colon cancer.

The incidence of colon and rectal cancer (CRC) is increasing in the population under 50 years of age, with more than 10% of cases occurring in young adults [4]. CRC with familiar predisposition (without genetic correlation), hereditary CRC (like hereditary non-polyposis colorectal cancer or adenomatosis polyposis syndrome), and chronic inflammatory bowel disease are associated with younger age [5,6,7,8].

CRC treatment is often based on multimodal therapies, including surgery, chemotherapy, radiotherapy, and, more recently, immunotherapy, which makes it difficult to estimate the expected effect of oncological treatment on clinically relevant gonadotoxicity. Advances in medical therapy have led to an increase of approximately 65% in 5-year survival rates for CRC for all tumor stages. For stage I, the 5-year survival is about 90% [9,10].

There is increasing awareness and knowledge regarding the toxicity of cancer treatments and long-term complications such as hormonal changes, uterine changes, and loss of ovarian function due to chemotherapy and radiotherapy, leading to infertility in a group of long-term survivors [11]. The standard adjuvant chemotherapy regime for stage II/III is the FOLFOX regime, which includes folinic acid, 5-fluorouracil, and capecitabine or oxaliplatin. Chemotherapy is often combined with radiotherapy in rectal cancer [12]. The reproductive toxicity of chemotherapy is estimated to be low to intermediate, but radiotherapy of the pelvis is presumed to substantially harm the gonads and uterus. This effect could be reduced by fertility preservation methods, such as the freezing of oocytes, transposition of the ovaries, or, as described recently, transposition of the uterus [13,14]. It is also assumed to be associated with several kinds of late toxicity, including gastrointestinal toxicity.

Therefore, fertility preservation has become more relevant. However, the handbook of the network FertiPROTEKT [15] and the ESHRE fertility preservation guideline [11] are among the very few sources that contain specific recommendations for fertility preservation in colon and rectal cancer. This might be due to the limited data on the gonadotoxicity of multimodal CRC treatment.

Counseling about fertility preservation measures is crucial before the start of gonadotoxic therapy. We, therefore, systematically analyzed the published literature on the gonadotoxic effects of CRC treatments in order to better counsel patients on the risk of infertility and the need for fertility preservation measures. This meta-analysis is part of the FertiTOX [16] project (www.fertitox.com), organized by FertiPROTEKT (www.fertiprotekt.com), which aims to fill the data gap on the gonadotoxicity of cancer therapies to enable more accurate counseling regarding fertility preservation [17,18,19,20].

## 2. Materials and Methods

### 2.1. Registration of Protocols

This study’s protocol has been registered in the Prospective International Registry of Systematic Reviews (PROSPERO; Registry Number: CRD42024511944). The Preferred Reporting Criteria for Systematic Reviews and Meta-Analyses (PRISMA) were used [21].

### 2.2. Search Strategy

We conducted a systematic literature search of Medline, Embase, the Cochrane database of systematic reviews, and CENTRAL in March 2024 (Figure 1). A specialized librarian developed an initial Embase search strategy and tested a basic reference list. Following refinement and query, complex search strategies were developed for each information source based on database-controlled vocabularies (thesaurus terms/headings) and text terms.

The text-word search included synonyms, acronyms, and similar terms. We limited our search to publications from 2000 to March 2024. Our search terms included all types of colorectal cancer.

A double-negative search strategy based on the Ovid “humans-only” filter excluded animal-only studies from the searches. The detailed final search strategies are provided in Appendix A. In addition to the electronic database search, reference lists and bibliographies of relevant publications were reviewed for relevant studies. All identified citations were imported into the software Covidence (https://www.covidence.org) accessed on 1 March 2024, a tool for systematic reviews. Duplicate records were removed [22].

### 2.3. Inclusion and Exclusion Criteria

Studies were independently assessed for inclusion using (www.covidence.org) [23] by four investigators (CA, AV, HH, and EP). All original articles that provided information on the colorectal cancer type, therapy, and fertility outcomes with numbers sufficient to calculate prevalence were included. Definitions of clinically relevant gonadal toxicity are described in Table 1. Studies in which gonadotoxicity could not be assessed using the criteria in Table 1 were excluded.

### 2.4. Data Extraction

Four investigators (CA, AV, HH, and EP) abstracted and then independently reviewed the data. Characteristics of the study populations (patient age at diagnosis and outcome, duration of follow-up, type of CRC, type of oncological treatment, and fertility parameters) were the principal variables of interest (Table 2 and Table 3). Discrepancies were discussed and resolved by consensus.

### 2.5. Quality Assessment

The Newcastle–Ottawa Scale (NOS) 17 was used to assess the quality of individual studies. The scoring of individual studies was based on three parameters: subject selection (0–4 stars), comparability (0–2 stars), and study outcome (0–3 stars). The scoring was as follows: good quality (=3 or 4 stars in selection AND 1 or 2 stars in comparability AND 2 or 3 stars in outcome/exposure), fair quality (=2 stars in selection AND 1 or 2 stars in comparability AND 2 or 3 stars in outcome/exposure), and poor quality (=0 or 1 star in selection OR 0 stars in comparability OR 0 or 1 stars in outcome/exposure). All included studies were reviewed by CA, AV, HH, and EP to independently assess the risk of bias; disagreements were resolved by consensus. Scoring was conducted according to the terms listed in Table 4.

### 2.6. Data Synthesis

The prevalence of clinically relevant gonadotoxicity in men and women with colorectal cancer after oncological therapy was the primary outcome of our systematic review. In most of the studies, just one parameter, like amenorrhea or low testosterone level, was used to define gonadotoxicity (Table 2 and Table 3). Subgroup analysis with chemotherapy alone, radiotherapy alone, and the combination of both types of treatment was performed. To calculate the prevalence, the number of patients who met the criteria for clinically relevant gonadotoxicity was divided by the number of patients at risk, as reported in the individual studies. For the pooled prevalence, statistical analyses were performed using the “metafor” function in the R software (Version 4.2.3, R Core Team, Vienna, Austria, 2013). Heterogeneity was assessed using Cohen’s Q statistic and I statistic2. In the presence of high heterogeneity, random-effects models were used. Studies with unspecified treatment were excluded from the outcome assessment to provide clinically meaningful estimates in the meta-analysis.

## 3. Results

### 3.1. Results of the Systematic Review

A total of 67 out of 4420 studies were included in the full-text analysis after screening 3581 abstracts (11 studies were presented twice by Covidence). The main reasons for excluding the 3514 abstracts were a lack of clear reference to clinically relevant gonadotoxicity or a lack of original data. Finally, 22 articles were included in the systematic review and meta-analysis (Figure 1). Forty-five studies were excluded because they did not meet the prespecified inclusion criteria.

### 3.2. Study Characteristics

The characteristics of the 22 studies are summarized in Table 2 and Table 3.

The included studies were retrospective (n = 10) and prospective (n = 12). The reviewed studies reported menstrual status, gonadal dysfunction, and hormonal changes as female clinically relevant gonadotoxicity outcomes. Male clinically relevant gonadotoxicity outcome parameters included sperm analysis and hormonal changes. The studies were performed with only men (12), only women (12), or both genders (2).

Except for five good-quality articles, the majority of studies (n = 17) were rated as being of poor methodological quality. This was mainly due to a lack of comparison groups (Table 4).

A total of 1634 patients were diagnosed with colorectal cancer (rectal and colon cancer, abbreviated as CRC) and underwent oncological treatment. A total of 775 (47.4%) women and 859 (52.6%) men were eligible for the analysis of clinically relevant gonadotoxicity. Rectal cancer was found in 1041 cases, of which 208 (20%) were female and 833 (80%) were male. Colon cancer was diagnosed in 163 cases, of which 145 (89%) were female and 18 (11%) were male. The corresponding numbers for CRC were 430, 422 (98.1%), and 8 (1.9%), respectively.

Study sample sizes ranged from 4 to 361 patients (4 to 361 in females and 8 to 290 in males). The studies were conducted in various regions, including Europe (n = 8), Asia (n = 9), and North America (5). One study analyzed patients with colon cancer, sixteen included patients with rectal cancer, and in six studies, the origin of cancer was not precisely defined (and therefore considered CRC in our study).

Study participants comprised post-pubertal males and females, with a median age of 34.5 years (range 18–50 years) in females and 56.2 years in males (range 20–87 years) at the time of cancer diagnosis. The age of the patients at the time of post-cancer fertility assessment was very different, as was the duration of the follow-up, ranging from 6 weeks to 12 years (mean 2.4 years, median 2 years).

Treatment options included surgery, chemotherapy, and radiotherapy.

### 3.3. Prevalence of Clinically Relevant Gonadotoxicity

The prevalence of gonadotoxicity in patients with a history of CRC ranged from 13% to 37% overall, from 11 to 54% in females, and from 13 to 26% in males. In retrospective studies of long-term survivors, the mean follow-up was 4.4 years in women [33], with a prevalence of clinically relevant gonadotoxicity of 75%, and 5 years in men [37], with a prevalence of gonadotoxicity of 16%.

## 4. Results of the Meta-Analysis

Ten studies [27,30,32,33,34,35,37,38,40,43] fulfilled the inclusion criteria and were considered for the meta-analysis.

### 4.1. Pooled Overall Prevalence of Gonadotoxicity After All Types of Treatment

Ten studies were eligible for inclusion in the analysis of the overall prevalence of clinically relevant gonadotoxicity. These studies comprised 615 female and 562 male cases. Consequently, patients were categorized according to their gender and oncological therapy (i.e., different types and doses of chemotherapy and radiotherapy and combinations of different therapies). The prevalence of gonadotoxicity in each of these studies and the data used in the meta-analysis are shown in Figure 2, Figure 3 and Figure 4. The prevalence of clinically relevant gonadotoxicity was 23% overall (95% CI: 13–37%), 27% (11–54%) in women, and 18% (13–26%) in men. The heterogeneity test revealed significant heterogeneity among the studies: I^2^ = 94%, *p* < 0.01; I^2^ = 96%, *p* < 0.01; and I^2^ = 60%, *p* 1.00.

### 4.2. Subgroup Analysis

The first subgroup analysis intended to evaluate gonadotoxicity based on the type of CRC (colon cancer vs. rectosigmoid). The analysis was only possible for rectosigmoid cancer (Figure 5). The prevalence of gonadotoxicity was 39% (95% CI: 20–64%) (Figure 5). Data heterogeneity was I^2^ = 95%, *p* < 0.01.

The second subgroup analysis intended to evaluate gonadotoxicity based on the type of cancer treatment (chemotherapy only, radiotherapy only, and the combination of both treatments). Three groups of treatments were evaluated (Figure 6, Figure 7 and Figure 8). The prevalence of clinically relevant gonadotoxicity in the chemotherapy-only group was 4% (95% CI: 2–10%) (Figure 6), and in the radiotherapy-only group, 23% (95% CI: 10–44%) (Figure 7). The prevalence of gonadotoxicity was found to be highest when chemotherapy and radiotherapy were combined, at 68% (95% CI: 40–87%) (Figure 8). The heterogeneity test revealed significant heterogeneity among the studies: I^2^ = 91%, *p* < 0.01; I^2^ = 89%, *p* < 0.01; and I^2^ = 0%, p 1.00, respectively.

## 5. Discussion

The aim of this systematic review and meta-analysis was to analyze the prevalence of clinically relevant gonadotoxicity after colorectal cancer in order to improve fertility counseling. To the best of our knowledge, this is the first meta-analysis of the overall prevalence of gonadotoxicity after multimodal oncological treatment for colorectal cancer.

Our review revealed the following results.

First, the overall pooled prevalence of clinically relevant gonadotoxicity in the general population of CRC survivors was 23% (95% CI: 13–37%). Based on the categorization of treatment-induced infertility, where <20% = low risk, 20–80% = intermediate, and >80% = high risk, 23% corresponds to a low to intermediate risk. When categorized by sex, the gonadotoxicity prevalence was 27% (95% CI: 11–54%) in women and 18% (95% CI: 13–26%) in men. Second, the subgroup analysis for rectosigmoid cancer showed a gonadotoxicity prevalence of 39% (95% CI: 20–64%), which corresponds to an intermediate risk, and third, the prevalence of gonadotoxicity was highest with the combination of radiotherapy and chemotherapy at 68% (95% CI: 40–87%), which corresponds to an intermediate to high risk, compared to radiotherapy alone at 23% (95% CI: 10–44%) or chemotherapy alone at 4% (95% CI: 2–10%).

We identified five retrospective studies that were of good quality [31,32,33,37,42].

A subgroup analysis for colon cancer was not possible due to mixed cohorts. These cohorts included different combinations and doses of chemotherapy, pooled results, and mixed-age populations.

Gonadotoxicity is associated with the age of female patients, the chemotherapy regimen, the cumulative dose of chemotherapy and radiotherapy, the type of surgery, and the patient’s reproductive status. It is important to note that cancer treatment tends to be more gonadotoxic in younger patients than in older patients with CRC [25,27,46,47].

Adjuvant FOLFOX (5-fluorouracil, leucovorin, and oxaliplatin) chemotherapy is the standard treatment for CRC.

The risk of treatment-related permanent amenorrhea in women and temporary reduction in sperm count in men caused by fluorouracil is very low [48,49].

Levi et al. (2015) observed the effects of oxaliplatin among 19 CRC patients (11 women and 8 men) who underwent an assessment of hormone levels before and six months after treatment [28]. In women, the anti-Müllerian hormone (AMH) concentration decreased and the follicle-stimulating hormone level increased, but all patients continued menstruating or resumed menstruation post-treatment. Some detrimental effects caused by oxaliplatin were also seen, as shown by slightly reduced inhibin B post-treatment [25].

Cercek et al. (2013) evaluated the incidence of FOLFOX-induced amenorrhea in female cancer survivors. The results showed that 16% of women had persistent amenorrhea after FOLFOX chemotherapy [25].

Our results show that the chemotherapy-induced risks are low to intermediate for colorectal cancer. However, the risk is intermediate to high if pelvic radiotherapy is added, as already shown by others [11,50]. For rectal cancer, high-dose pelvic radiotherapy is a standard treatment. In women, doses of less than 2 Gy have been observed to reduce the number of immature oocytes in the ovaries by 50% [33,51,52]. More than 90% of patients with rectal cancer receive radiation doses of 45–50 Gy, causing premature ovarian failure [53].

Radiotherapy may also affect fertility by damaging the uterus or testes.

High doses of radiation can cause uterine infertility in women. In adults, whole-body irradiation with doses of 12 Gy is associated with an increased risk of miscarriage, premature birth, and low-birth-weight infants. Irradiation of the uterus with doses > 25 Gy in childhood or >45 Gy in adults leads to uterine infertility, and patients should be advised not to become pregnant [54].

Looking at the male side, radiotherapy for CRC causes damage to the testis, as shown by increased gonadotropin levels and decreased testosterone levels, with a risk of permanent infertility and endocrine failure [55]. Long-lasting azoospermia can be expected if the testicles are exposed to ≥2 Gy, and permanent azoospermia is possible if the exposure is ≥4 Gy [15].

In recent years, immunotherapy has been introduced into the treatment of CRC. Neoadjuvant immunotherapy may improve the therapy of patients with microsatellite-instable colorectal cancer undergoing chemotherapy [56,57]. The anti-VEGF (vascular endothelial growth factor) antibody bevacizumab, so far, has an unknown risk of treatment-related infertility [50].

Previous studies have shown that surgery, especially low anterior resection (LAR) for rectal cancer, can cause neurological dysfunction and therefore affect bowel and bladder function, sexuality, and fertility [58,59]. In men, neurological dysfunction due to surgery can also affect ejaculation, which further reduces fertility. The presence of a stoma is also associated with poorer sexual function [60,61].

All treatment options for colorectal cancer have the potential to damage gonadal function to varying degrees; therefore, we suggest considering the results of this study, potentially on an individual treatment basis, for the choice of future treatments. Therefore, fertility preservation should always be considered if a high risk of infertility due to surgery, chemotherapy, or radiotherapy is expected. This is in line with a growing awareness of survivorship care among long-term survivors of colorectal cancer [62].

Although we strictly followed the recommendations for producing high-quality evidence summaries, there are some limitations to our study: First, the majority of the included studies were based on retrospective data, which did not provide the necessary information on the long-term effects on clinically relevant gonadotoxicity. Second, the heterogeneity of the treatment and study populations precluded additional subgroup analyses. Such subgroup data would be relevant for individualized fertility preservation counseling. Finally, the short follow-up period did not allow an assessment of the long-term effects of cancer therapy on fertility. Another limitation may be the fact that some of the patients may already have had a gonadal dysfunction of some kind before the treatment. We also consider the males’ very old average age of 56.2 years to be a limitation of the studies. This does not correspond with the average male reproductive age.

## 6. Conclusions

In conclusion, this first meta-analysis evaluates the pooled prevalence of clinically relevant gonadotoxicity after CRC treatment. It provides clinically relevant information to counsel patients about the risk of infertility and the need to consider fertility preservation measures. All fertile patients with colorectal cancer should be aware of the risk of therapy to their fertility. This meta-analysis delivers a basis to advise all patients with colorectal cancer. The prevalence of gonadotoxicity was low in the case of chemotherapy only but rather high in the case of radiotherapy or radiochemotherapy. However, fertility preservation is also recommended in chemotherapy-only cases because dose-intensive follow-up treatments cannot be excluded and because extensive, longitudinal data on individual treatment effects are lacking. To date, no meta-analysis has summarized and analyzed the current literature on the fertility aspect of colorectal cancer in order to advise patients regarding fertility preservation. It is important that young patients suffering from colorectal cancer receive adequate consultation, including on the risk of gonadotoxicity from the planned treatment, so that they still have the chance to plan a family. Further prospective studies are needed to establish the individual impact of CRC treatment on gonadal function and to evaluate the effect of new treatment modalities, such as immunotherapies.

## Figures and Tables

**Figure 1 cancers-16-04005-f001:**
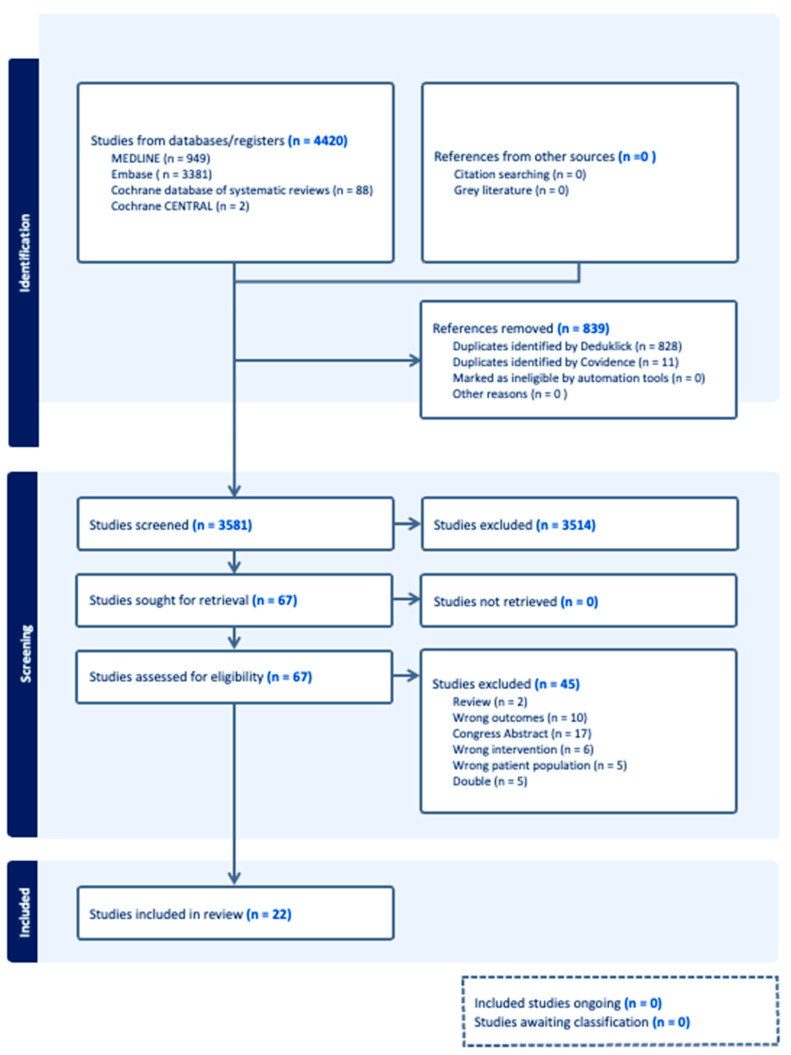
PRISMA flow diagram. A flowchart of the literature search and selection process.

**Figure 2 cancers-16-04005-f002:**
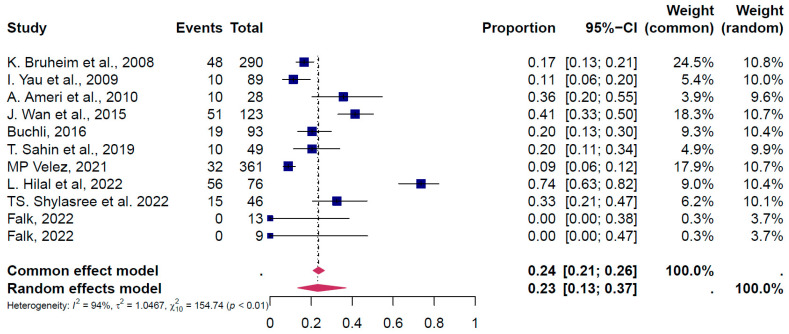
The pooled overall prevalence of general gonadotoxicity [27,30,32,33,34,35,37,38,40,43]. A forest plot of the proportions and 95% confidence intervals (CIs) in studies that evaluated the prevalence of clinically relevant gonadotoxicity in women and men following gonadotoxic therapy for CRC, where 0 means 0% clinically relevant gonadotoxicity and 1 = 100% clinically relevant gonadotoxicity. The blue square for each study indicates the proportion, the size of the box indicates the weight of the study, and the horizontal line indicates the 95% CI. The data in bold and the pink diamond represent the pooled prevalence for post-treatment clinically relevant gonadotoxicity and 95% CI. Overall estimates are shown in the fixed- and random-effects models.

**Figure 3 cancers-16-04005-f003:**
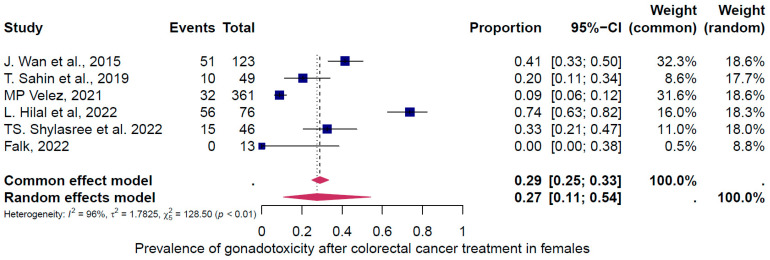
Pooled overall prevalence of gonadotoxicity in women [27,30,32,33,34,35]. For details, see legend of Figure 2.

**Figure 4 cancers-16-04005-f004:**
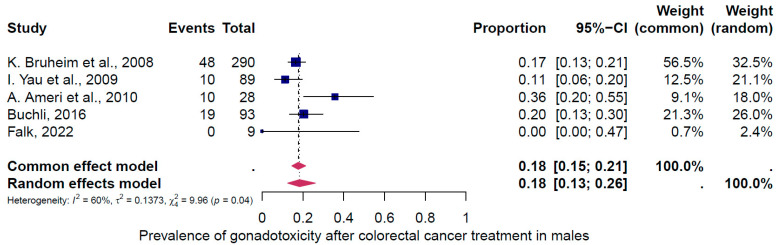
Pooled overall prevalence of gonadotoxicity in men [35,37,38,40,43]. For details, see legend of Figure 2.

**Figure 5 cancers-16-04005-f005:**
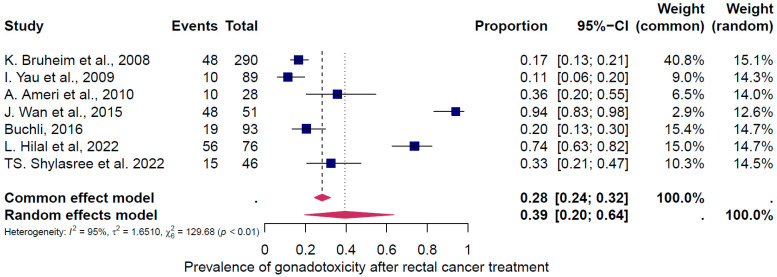
Pooled overall prevalence of gonadotoxicity, subgroup for rectal cancer [27,33,34,37,38,40,43]. For details, see legend of Figure 2.

**Figure 6 cancers-16-04005-f006:**
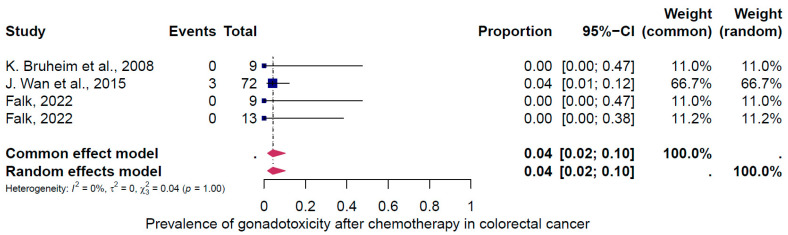
Pooled overall prevalence of gonadotoxicity among those who received chemotherapy only [27,35,37]. For details, see legend of Figure 2.

**Figure 7 cancers-16-04005-f007:**
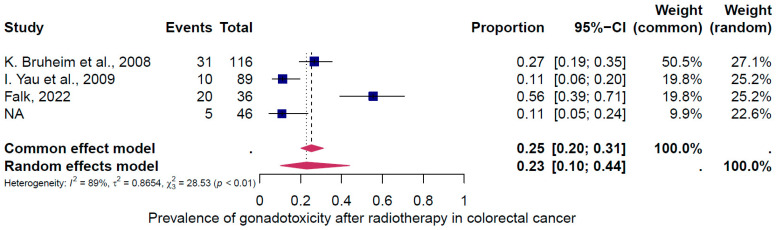
Pooled overall prevalence of gonadotoxicity among those who received radiotherapy only [35,37,38]. For details, see legend of Figure 2.

**Figure 8 cancers-16-04005-f008:**
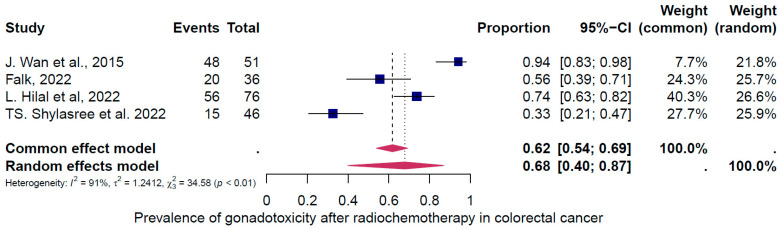
Pooled overall prevalence of gonadotoxicity among those who received the combination of radiotherapy and chemotherapy treatment [27,33,34,35]. For details, see legend of Figure 2.

**Table 1 cancers-16-04005-t001:** Definitions of clinically relevant gonadotoxicity.

Females	Males
Menstrual cycle disorders Amenorrhea/oligomenorrhea Hormonal treatment: puberty induction/hormonal replacement therapy	Disorders of sperm quality Azoospermia Oligozoospermia
Hormone levels above the normal range Follicle-stimulating hormone (FSH) Luteinizing hormone (LH)	Hormone levels above the normal range Follicle-stimulating hormone (FSH) Luteinizing hormone (LH)
Premature ovarian insufficiency (POI) Oligo-/amenorrhea for at least 4 months and an elevated FSH level > 25 IU/L on two occasions at 4 weeks apart before the age of 40. (ESHRE Definition)	Gonadal dysfunction Low testosterone levels Hormonal treatment: testosterone therapy
Low ovarian reserve parameters Anti-Müllerian hormone (AMH) not detectable	Hormone levels below the normal range Inhibin B

**Table 2 cancers-16-04005-t002:** Characteristics of the included studies—females. Summary of cohort studies assessing the prevalence of gonadotoxicity in women.

First Author, Year of Publication	Country	Study Design	Number of Participants of Interest (Females)	Age of Participants of Interest at Time of Diagnosis/Therapy (Years, Range)	Age (Years, Mean ± SD) at Outcome/Evaluation	Follow-Up After Diagnosis/Treatment, Length in Years (Range)	Tumor Type Number (%)	Chemotherapy, Details	Radiotherapy, Details	Suspected Infertility	Comments
Al-Badawi et al., 2010 [24]	Saudi Arabia	Retrospective	4	23 (18–36)	Not specified	2.67 (0.83–5)	RC	Not specified	Yes, without specifications	2/4 (50%)	Calculated in women with persistent amenorrhea. Laparoscopic ovarian transposition to paracolic gutters with uterine conservation.
Cercek et al., 2013 [25]	USA	Retrospective	49	31–35 (21–50)	Not specified	>0.5 (range not specified)	CRC	FOLFOX standard modified mFOLFOX	No	8/49 (16%)	Calculated in women with persistent amenorrhea (>1 year).
Barahmeh et al., 2013 [26]	Jordan	Retrospective	4	Not specified	Not specified	3.5 (2.83–4.17)	RC	5-FU concomitantly with radiotherapy	Estimated irradiation dose to both ovaries after pelvic radiotherapy: 2.1 Gy for three patients and 18 Gy for one patient. External pelvic irradiation (45–60 Gy)	1/4 (25%)	Calculated in women with hypergonadotropic hypogonadism. Bilateral ovarian transposition to the paracolic gutter.
Wan et al., 2015 [27]	China	Retrospective	123	CC: 36 (17–40) RC: 35 (24–40)	Not specified	CC: 3.16 (1.52–6.32) RC: 3.35 (1.21–6.36)	CC 58.6 RC 41.4	FOLFOX XELOX Capecitabine only	CC: no RC: intensity-modulated radiotherapy to pelvis (total dose 45–55 Gy in 25–30 fractions)	colon cancer 3/72 (4.2%) rectal cancer 48/51 (94.1%)	Calculated in women with persistent amenorrhea > 1 year.
Levi et al., 2015 [28]	Israel	Prospective	11	36	36.5	0.5	CRC	FOLFOX or XELOX	In 1 patient	0/11 (0%)	Calculated in women with hypergonadotropic hypogonadism.
Sioulas et al., 2017 [29]	USA	Retrospective	22	39 (26–45)	Not specified	2.42 (0.09–6)	RC (90.9) AC (9.1)	FOLFOX CAPOX FOLFOX/bevacizumab FOLFOX/FOLFIRINOX Capecitabine 5-FU Mitomycin C	RC: 5000 to 5400 cGy to the rectal tumor 4500 cGy to the pelvic nodes AC: 5600 cGy to the primary tumor 4500 cGy to the pelvic nodes	6/18 (33.3%)	Calculated in women with hypergonadotropic hypogonadism. Only 18 patients were evaluable for ovarian function. Nineteen patients underwent OT.
Sahin et al., 2019 [30]	Turkey	Retrospective	60	40 (19–50)	Not specified	Min. 1	CC	5-FU alone 5-FU + oxaliplatin FOLFOX CAPOX	No	10/49 (20.4%)	Calculated in women with persistent amenorrhea >1 year.
Svanström Röjvall, 2020 [31]	Sweden	Prospective	6	Not specified	Not specified	2	RC	Yes	Short course (5 Gy × 5) Long course (2 Gy × 25 or 1· 8 Gy × 28) + 3 fractions of boost	5/6 (83.3%)	Calculated in women with undetectable AMH.
Velez, 2021 [32]	Canada	Retrospective	361	Not specified	Not specified	Not specified	CRC	Not specified	Not specified	32/361 (8.9%)	Calculated in women with infertility diagnosis using the health administrative database.
Hilal et al., 2022 [33]	USA	Retrospective	76	43 (20–49)	Not specified	4.48 (0.48–15.44)	RC	FOLFOX/XELOX 5FU/LV Xeloda Cisplatin–Etoposide	Median dose: 50 Gy (25–56) 25 (5–28) fractions 3D-CRT IMRT	56/76 (75%)	Twenty-six (34%) underwent OT. Calculated in women with hypergonadotropic hypogonadism.
Shylasree, 2022 [34]	India	Retrospective	46	25.2	Not specified	3.5 (0.42–6.75)	RC	Capecitabine 5-FU + oxaliplatin	Neoadjuvant chemoradiation: 50.4 Gy in 28 fractions (1.8 Gy) with concurrent capecitabine. Short-course RT: 25 Gy in five fractions (5 Gy).	15/43 (34.9%)	Calculated in women with hypergonadotropic hypogonadism and a need for puberty induction.
Falk, 2022 [35]	Norway, Sweden, Finland	Prospective	16	35 (range 20–40)	Not specified	1–5	CC RC AA CRC	FOLFOX CAPOX Nordic FLOX	No	0/13 (0%)	Calculated in women with hypergonadotropic hypogonadism, amenorrhea, and undetectable AMH.

Note: The studies are sorted by year of publication. Age and duration of follow-up are given as years with mean (SD) or with range where such data are available. Abbreviations: Diagnosis: CRC = colorectal cancer; CC= colon cancer; RC = rectal cancer; AC = anal cancer; AA = appendiceal adenocarcinoma. Chemotherapy: FOLFOX = 5-fluorouracil, leucovorin [folinic acid], and oxaliplatin; XELOX = capecitabine and oxaliplatin; CAPOX = capecitabine and oxaliplatin; Nordic FLOX = 5-FU bolus, folic acid, and oxaliplatin; 5-FU = 5-fluorouracil; LV = leucovorin. Radiotherapy: Parameters: AMH: anti-Müllerian hormone. Other: OT = ovarian transposition.

**Table 3 cancers-16-04005-t003:** Characteristics of the included studies—males. Summary of cohort studies assessing the prevalence of gonadotoxicity in men.

First Author, Year of Publication	Country	Study Design	Number of Participants of Interest (Males)	Age of Participants of Interest at Time of Diagnosis/Therapy	Age, yrs (Mean ± SD) at Outcome/ Evaluation	Follow-Up After Diagnosis/ Treatment, Length in Years (Range)	Tumor Type	Chemotherapy, Details	Radiotherapy, Details	Suspected Infertility (…/…/%) MALES	Comments
Piroth et al., 2003 [36]	Germany	Prospective	18	not specified	Not specified	Not specified	RC	5-FU	Total dose: 50.4 Gy Single dose: 1.8 Gy per day 5 × per week TD: Mean: 0.057 Gy (0.035–0.114) Cumulative: 1.60 Gy (0.98–3.19)	n/a	
Bruheim et al., 2008 [37]	Norway	Retrospective	290	irradiated 66.0 (45.1–86.0) non-irradiated 71.4 (40.2–94.8)	Not specified	2–12	RC	5-FU + leucovorin	Mean dose: 50.07 Gy (25 fractions of 2 Gy given in 5 weeks) Treatment time: 35 days (7–106) Preoperative: 74 (63.8%) Postoperative: 42 (36.2%)	48/290 (16.6%)	Calculated in men with testosterone values under the normal limit
Yau et al., 2009 [38]	Canada	Prospective	89	EBRT 62.25 (32–87) HDRBT 61.03 (37–84)	Not specified	1.42 - 1.17 EBRT - 1.67 HDRBT	RC	5-FU	EBRT (38 patients) 45.0–50.4 Gy in 1.8 Gy per day (5 days per week over 5–5.5 weeks) HDRBT (51 patients) 26 Gy (4 times per day; 6.5 Gy daily) TD: - EBRT: 1.24 Gy (0.06–7.80) - HDRBT: 0.27 Gy (0.14–0.65)	EBRT 9/51 (17.6%) HDRBT 1/38 (2.6%) total 10/89 (11.2%)	2-year hypogonadism rates
Yoon et al., 2009 [39]	England	Prospective	43	56.5 (35–72)	Not specified	6,1 (1.3–9.4)	RC	Adjuvant: 5-FU (bolus) Concurrent: - CIVI (36; 84%) - bolus (7; 16%) 2 additional 5-day cycles of 5-FU (450 mg/m^2^/d)	Median dose: 54.0 Gy in 30 fractions TD: 4 Gy (1.5–8.9 Gy) Three-field pelvic technique (36, 84%) Four-field technique (5, 11.6%)	Only mean values	
Ameri et al., 2010 [40]	Iran	Prospective	28	52.72 ± 13	Not specified	0.13	RC	Adjuvant 18 (Co60: 10 LINAC: 8) Neo-adjuvant 6 (Co60: 2 LINAC: 4) Palliative 1 (Co60) 5-FU (CIVI) (Co60: 4 LINAC: 1) 5-FU + oxaliplatin (Co60: 3 LINAC: 1) Capecitabine (Co60: 5 LINAC: 8)	Co60 (14 patients) 47.88 Gy ± 2.77 LINAC (14 patients) 47.55 Gy ± 3.24 TD: Co60 (4) 55 mGy (±24.7) (29–80) Mean cumulative: 3.27 Gy (2.4–3.8) 6.6% (4.7–7.5%) of total target dose LINAC (5): 120 mGy (±20.3) (85–135) Mean cumulative: 1.4 Gy (0.73–2) 3% (1.6–4.45) of total target dose	(10/28) 35.71%	Of patients with a decrease in testosterone post-radiotherapy
Hennies et al., 2012 [41]	Germany	Prospective	83	65 (39–83)	Not specified	1	RC	Concomitant: 5-FU (53, 64%) 5-FU + oxaliplatin (30, 36%) Adjuvant: 5-FU (68, 88%) 5-FU + oxaliplatin (9, 12%)	Isocentric three-field posterior–anterior/lateral technique Total dose: 50.4 Gy (1.8 Gy daily, 5 days/week) TD: 3.9 Gy	Only mean values	
Buchli et al., 2015 [42]	Sweden	Prospective	40	59.9 ± 12.8	Not specified	1	RC	Postoperative chemotherapy (12/40 patients)	Preoperative radiotherapy: short-course (5 × 5 Gy) (30/40 patients) 28 × 1.8 Gy (10/40 patients)	6/40 (15%)	Calculated in men with testosterone values under the normal limit
Levi et al., 2015 [28]	Israel	Prospective	8	38 (33–41)	38.5	0.5	CRC	FOLFOX XELOX	n/a	none	
Buchli et al., 2016 [43]	Sweden	Prospective	105	60.3 (±11.3)	60.3 (±11.3)	0.1 (0.01–0.53)	RC	Concomitant chemotherapy (23/25) with long-course RT Full-dose preoperative chemotherapy (11/68) with short-course RT	Preoperative RT: 25 Gy (short-course RT, 5 Gy × 5) or 50.4 Gy (long-course RT, 1.8 Gy × 28) Full-dose preoperative chemotherapy: after short-course RT according to the protocol of the RAPIDO trial	n/a	
Motte et al., 2021 [44]	Sweden	Prospective	115	Group A: 52 Group B: 63	Not specified	2	RC	Capecitabine, 5-FU, oxaliplatin, leucovorin, irinotecan	TD: Group A: 2.6% Group B: 1.8%	(5/8) 62.5%	Patients with oligospermia 2 years after therapy Group A = semen sample Group B = no semen sample
Falk et al., 2022 [35]	Norway Sweden Finland	Prospective	20	35 (20–40)	Not specified	1–5	CC (90%) RC (10%)	CAPOX Nordic FLOX (17, 85%) FOLFOX/FLOX CAPOX	No radiotherapy	0/9 (0%)	Calculated in men with normal FSH/LH
Krishna et al., 2022 [45]	India	Prospective	20	59.5	Not specified	0.1	RC	Concurrent: capecitabine 825 mg/m^2^ (2x per day, five days a week, along with radiation)	3DCRT (6, 30%) IMRT (14, 70%) neoadjuvant (5, 33%) adjuvant (15, 67%) 50.4 Gy for 5 weeks delivered in 28 fractions TD: 2.65 Gy (1.96 Gy to 4.96 Gy) 5.25% of the total dose	5/20 (25%)	Calculated in men with testosterone values under the normal limit

Note: The studies are sorted by year of publication. Age and duration of follow-up are given as years with mean (SD) or with range where such data are available. Abbreviations: General: TD = Testicular Dose; CIVI = continuous intravenous infusion. Diagnosis: CRC = colorectal cancer; CC= colon cancer; RC = rectal cancer; AC = anal cancer; AA = appendiceal adenocarcinoma. Radiotherapy: EBRT = external beam radiotherapy; HDRBT = high-dose-rate brachytherapy; 3DCRT = 3-dimensional conformal radiation therapy; IMRT = intensity-modulated radiation therapy; Co60 = Cobalt 60; LINAC = linear accelerator. Chemotherapy: FOLFOX = 5-fluorouracil, leucovorin [folinic acid], and oxaliplatin; XELOX = capecitabine and oxaliplatin; CAPOX = capecitabine and oxaliplatin; Nordic FLOX = 5-FU bolus, folic acid, and oxaliplatin; 5-FU = 5-fluorouracil; LV = leucovorin. Parameters: AMH: anti-Müllerian hormone.

**Table 4 cancers-16-04005-t004:** Bias screening. Newcastle–Ottawa Quality Assessment form for Cohort Studies.

	Selection				Comparability	Outcome					
First Author, Year of Publication	Representativeness of Exposed Cohort	Selection of Non-exposed Cohort	Ascertainment of Exposure	Outcome of Interest Not Present at Study Start	Comparability of Cohorts on the Basis of the Design or Analysis Controlled for Confounders	Assessment of Outcome	Sufficient Length of Follow-Up for Outcomes to Occur	Adequacy of Follow-Up of Cohorts	Total	Quality Assessment	Comments
Piroth et al., 2003 [36]	★	-	★	★	-	-	-	★	4/8	poor	no non-exposed cohort group
Bruheim et al., 2008 [37]	★	★	★	-	★	★	★	-	6/8	good	
Yau et al., 2009 [38]	★	-	★	★	-	★	★	★	6/8	poor	no non-exposed cohort group
Yoon et al., 2009 [39]	★	-	★	★	-	★	★	★	6/8	poor	no non-exposed cohort group
Al-Badawi et al., 2010 [24]	★	-	★	-	-	★	★	★	5/8	poor	no non-exposed cohort group
Ameri et al., 2010 [40]	★	-	★	★	-	★	-	★	5/8	poor	no non-exposed cohort group
Hennies et al., 2012 [41]	★	-	★	★	-	★	★	★	6/8	poor	no non-exposed cohort group
Barahmeh et al., 2013 [26]	★	-	★	★	-	★	★	★	6/8	poor	no non-exposed cohort group
Cercek et al., 2013 [25]	★	-	★	-	-	-	★	★	4/8	poor	no non-exposed cohort group
Buchli et al., 2015 [42]	★	-	★	★	-	★	★	★	6/8	poor	no non-exposed cohort group
Levi et al., 2015 [28]	★	-	★	★	-	★	★	★	6/8	poor	no non-exposed cohort group
Wan et al., 2015 [27]	★	-	★	-	-	★	★	-	4/8	poor	no non-exposed cohort group
Buchli et al., 2016 [43]	★	★	★	★	★	★	-	★	7/8	good	
Sioulas et al., 2017 [29]	★	-	★	-	-	-	★	★	4/8	poor	no non-exposed cohort group
Sahin et al., 2019 [30]	★	-	★	★	-	-	★	★	5/8	poor	no non-exposed cohort group
Svanström Röjvall et al., 2020 [31]	★	★	★	★	★	★	★	★	8/8	good	
Motte et al., 2021 [44]	★	-	★	★	-	★	★	-	5/8	poor	no non-exposed cohort group
Velez et al., 2021 [32]	★	★	★	-	★	★	★	★	7/8	good	
Falk et al., 2022 [35]	★	-	★	★	-	★	★	★	6/8	poor	no non-exposed cohort group
Hilal et al., 2022 [33]	★	★	★	-	★	★	★	★	7/8	good	
Krishna et al., 2022 [45]	★	-	★	★	-	★	-	★	5/8	poor	no non-exposed cohort group
Shylasree et al., 2022 [34]	★	-	★	★	-	-	★	★	5/8	poor	no non-exposed cohort group

Star: yes.

## Data Availability

All the data utilized in the study are publicly available and/or contained within the manuscript.

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
