# Peer review of "Long-Term Effects on Gonadal Function After Treatment of Colorectal Cancer: A Systematic Review and Meta-Analysis"

_cancers, 2024, doi:10.3390/cancers16234005_

Round 1

Reviewer 1 Report

Comments and Suggestions for Authors

Report on cancers-3275707

Authors of the submitted manuscript examined the published literature on the gonadotoxic effects of CRC treatments to better understand the consequences of different therapies and to advice patients on the risk of infertility and the need for fertility preservation measures. Authors properly analyzed the literature and performed meta-analysis of eligible data, reporting on study details, inclusion/exclusion criteria and limitations. Since the knowledge concerning possible secondary effects of CRC therapy on gonadal function is scarce, additional work on the subject is welcome.

The paper is clearly and concisely written and statements supported by data and references. The obtained results suggested which treatment seems to have the greatest negative consequences.

Before accepting the manuscript, here are some suggestions to improve the article.

Table 1. defines the outcomes relevant to diagnose gonadotoxicity and Tables 2 and 3 point (in Comments) how was gonadotoxicity established in each study. Could you clarify in text the number of studies that made conclusions based on more than one determined parameter, as it seems that majority of consequences were named after diagnosis of deviation of just one parameter. Furthermore, is there a knowledge on the gonadal status of patients both before treatment for CRC and illness itself? Baseline data for patients are missing in Tables 2 and 3, so it is possible that some of them had gonadal dysfunction of certain kind even before. Some comment on this issue (at least as limitation) is needed if data are actually not available.

In Table 3, column „Suspected infertility...“ reports „n/a“, „only mean values“ or „none“ for some studies and without additional comment. How was, then, infertility diagnosed and considered?

Women were, generally, younger than men. So, women seem to be more prone to CRC in younger age than men and/or older males present to doctors latter in life. The mean age of male patients was 56.2 years and it is visible from Table 3. that subjects included in many studies are mostly older – they are not exactly in their reproductive period of life although there are no formal limits. At that age, a certain degree of decline in gonadal function is natural/expected. Could you analyse the prevalence of gonadal indicators for younger men, as that would more reliably document how actual is the problem?

Tables 2 and 3 should be mentioned in section 2.4.

Comments in Table 2 have one, two and three asterix, but two and three are not present in any other column.

In section 3.2., it is written „The studies were accomplished either with only men (12), only women (12), or both genders (2) - previously it was said that 22 studies were included.

Author Response

Editor-in-Chief 

Prof. Dr. Samuel Mok

Department of Gynecologic Oncology and Reproductive Medicine

The University of Texas MD Anderson Cancer Center

Houston, TX 77030, USA

Cancers

Zürich, 04.11.2024

Dear members of the editorial board, Dear Editors-in-Chief,

We thank the reviewer for taking the time and reading our manuscript "Long-term effects on gonadal function after treatment of colorectal cancer: A systematic review and meta-analysis» and for their helpful comments and suggestions that have helped us to considerably improve the quality of our article.

We have addressed all the points raised and modified the manuscript accordingly; the revisionsare highlighted in yellow and red.

Please find a detailed point-by-point response below.

Comments 1:

Table 1. defines the outcomes relevant to diagnose gonadotoxicity and Tables 2 and 3 point (in Comments) how was gonadotoxicity established in each study. Could you clarify in text the number of studies that made conclusions based on more than one determined parameter, as it seems that majority of consequences were named after diagnosis of deviation of just one parameter. Furthermore, is there a knowledge on the gonadal status of patients both before treatment for CRC and illness itself? Baseline data for patients are missing in Tables 2 and 3, so it is possible that some of them had gonadal dysfunction of certain kind even before. Some comment on this issue (at least as limitation) is needed if data are actually not available.

Our answer:

Thank you for pointing this out. We agree with this comment. In most of the studies there war just one parameter like amenorrhoe or low testosterone level to define gonadotoxicity. We just added this last sentence on page 10 2.6 data synthesis line 190-191. In females there was only one study (Falk et al.) with more than one paramter. In males there was just one parameter to define gonadotoxicity.

Unfortunately there was no knowledge on gonadal status of patients before treatman and illeness itself. As you mentions it is abolutely possible or better realistic, that some of them had gonadal dysfunction of certain kind before. Therefore we added a sentence on the discussion on page 16 line 377-378.

Comments 2:

In Table 3, column „Suspected infertility...“ reports „n/a“, „only mean values“ or „none“ for some studies and without additional comment. How was, then, infertility diagnosed and considered?

Our answer:

Thank you for your comment. In this study it is described that the fertility was reduced, but it was not clearly described with percentages. We found this information to be relevant, but we could exclude the study if you wish to.

Comments 3:

Women were, generally, younger than men. So, women seem to be more prone to CRC in younger age than men and/or older males present to doctors latter in life. The mean age of male patients was 56.2 years and it is visible from Table 3. that subjects included in many studies are mostly older – they are not exactly in their reproductive period of life although there are no formal limits. At that age, a certain degree of decline in gonadal function is natural/expected. Could you analyse the prevalence of gonadal indicators for younger men, as that would more reliably document how actual is the problem? 

Our answer:

Thank you for pointing this out. It is absolutely right that there is a great gap comparing the age of women and. men described in the studies. We think, that women are not more prone to CRC in younger age then men, but the studies examined women in their fertile phase of live bevor menopause compared to all men due to the fact, that men don`t have such a strict limitation of fertility like women. Unfortunately there were no data of younger men available. We added a few sentences according to this limitation on page 16 in the discussion section line 379-380: We also consider the males very old average age of 56,2 years as a limitation of the studies. This does not correspond with the average male reproductive age.

Comments 4:

Tables 2 and 3 should be mentioned in section 2.4.

Our answer:

Thanks a lot for this comment. We added this on line 152.

Comments 5:

Comments in Table 2 have one, two and three asterix, but two and three are not present in any other column.

Our answer:

We appreciate your attention to detail. We have corrected it.

Comment 6:

In section 3.2., it is written „The studies were accomplished either with only men (12), only women (12), or both genders (2) - previously it was said that 22 studies were included.

Our answer:

Thank you for pointing this out. There are studies which include only male, only women or both genders, so that we don’t have the addition of 26 studies, because some study include both men and women. That’s why the 22 studies are correct. 

Sincerely,

Dr. med. Christiane Anthon

OVA IFV, Clinic Zurich, Zurich, Switzerland

E-Mail: Christiane.wachter@gmx.de

Reviewer 2 Report

Comments and Suggestions for Authors

Dear colleague, i apreciate the enfortir that suposes studies like the one you have done. 
CRC and the effects of their trestment are becoming very important even fot he yuong people that afects . 
i think in your paper we need some aclarations: 

1. The study was approved for some Ethcis Committee?

2. Rectal and colon càncer are mixed Why? I think there are diferent tapes of cancer and the manegement are diferent too

3. what are the limitations of the study

4. what are the global conclusions and your recomendations ?

congratulations 

Author Response

Editor-in-Chief 

Prof. Dr. Samuel Mok

Department of Gynecologic Oncology and Reproductive Medicine

The University of Texas MD Anderson Cancer Center

Houston, TX 77030, USA

Cancers

Zürich, 04.11.2024

Dear members of the editorial board, Dear Editors-in-Chief,

We thank the reviewer for reading our manuscript "Long-term effects on gonadal function after treatment of colorectal cancer: A systematic review and meta-analysis» and for their helpful comments and suggestions that have helped us to considerably improve the quality of our article.

We have addressed all the points raised and modified the manuscript accordingly; the revisionsare highlighted in yellow.

Please find a detailed point-by-point response below.

Comments 1:

The study was approved for some Ethcis Committee?

Thank you for your question. Our study did not require Ethics Committee approval, as it is a systematic review. We conducted the work following the guidelines outlined by PROSPERO (Registry Number: CRD42024511944).

Comments 2:

Rectal and colon cancer are mixed Why? I think there are diferent tapes of cancer and the manegement are diferent too

Thank you for highlighting this important distinction. We agree that colon and rectal cancers are distinct types with notable differences in their biological behavior, location, and clinical management. However, in line with international guidelines and the studies we reviewed, these cancers are frequently grouped together under the umbrella of colorectal cancer (CRC). This combined classification allows for broader analysis of treatment impacts and outcomes, especially as certain treatment approaches—such as chemotherapy—overlap between the two. Nevertheless, we recognize that this approach may limit insights into the unique aspects of each cancer type, and we acknowledge this as a limitation in our discussion.

Comments 3:

what are the limitations of the study

Thank you for this comment, we described the limitations of our study on page 17 in the discussion line 373-383:

Although we strictly followed the recommendations for producing high-quality evidence summaries, there are some limitations to our study: First, the majority of the included studies were based on retrospective data, which did not provide the necessary information on the long-term effects on clinically relevant gonadotoxicity. Second, the heterogeneity of the treatment and study populations precluded additional subgroup analyses. Such subgroup data would be relevant for individualised fertility preservation counselling. Finally, the short follow-up period did not allow an assessment of the long-term effects of cancer therapy on fertility. Another limitation may be the fact that some of the patients may already have a gonadal dysfunction of a certain kind before the treatment. We also consider the males very old average age of 56,2 years as a limitation of the studies. This does not correspond with the average male reproductive age.

Comments 4:

what are the global conclusions and your recomendations ?

Thank you for this comment. We made global conclusions and recommendations in the discussion section page 17 line 384-395:

It provides clinically relevant information to counsel patients about the risk of infertility and the need to consider fertility preservation measures. The prevalence of gonadotoxicity was low in case of chemotherapy only but rather high in case of radiotherapy or radiochemotherapy. However, fertility preservation is also recommended in chemotherapy only cases because dose-intensive follow-up treatments cannot be excluded and because extensive, longitudinal data on individual treatment effects are lacking. Further prospective studies are needed to establish the individual impact of CRC treatment on gonadal function and to evaluate the effect of new treatment modalities, such as immunotherapies.

Sincerely,

Dr. med. Christiane Anthon

OVA IFV, Clinic Zurich, Zurich, Switzerland

E-Mail: Christiane.wachter@gmx.de

Reviewer 3 Report

Comments and Suggestions for Authors

In the current scenario of colorectal cancer with an increasingly younger incidence, this systematic review and meta-analysis analyzed the gonadotoxic effect and subgroup analyses were conducted according to gender, tumor location and regimen. The literature included was comprehensive and the clinical outcome was clearly defined, which provided preliminary evidence for fertility preservation. However, the following issues still require further clarification:

Main issues:

1. The results of this study suggested that all treatments had the potential to damage gonadal function, albeit to varying degrees, and this is consistent with our common sense, so I think that in the Discussion section, more attention should be paid to the possible implications of the results of this study for the choice of future treatments. The clinical significance of the present study is not sufficiently persuasive in the current article.

2. Have all included patients undergone surgery? Was the treatment analyzed in the article preoperative or postoperative? Surgery is also an important tool in the treatment of colorectal cancer, so why was the impact of surgery not considered?

3. Is it possible to conduct subgroup analyses based on age, as reproductive function, especially hormone levels, may change with age? Also, did the tumor staging have an impact on gonadal function?

Minor issues:

1. The reason for the exclusion of 3514 studies in the screening process was provided in the Results section of the manuscript, and it could be clearer by adding it to Figure 1.

2. For the convenience of reading, it is suggested to adjust the layout format of Tables 2 and 3, and to classify the outcomes of each study according to the contents of Table 1.

3. There are some clerical errors in the manuscript, such as in Lines 195-196, “The studies were accomplished either with only men (12), only women (12), or both genders (2)”, where the sum of the total studies exceeded 22.

Author Response

Editor-in-Chief 

Prof. Dr. Samuel Mok

Department of Gynecologic Oncology and Reproductive Medicine

The University of Texas MD Anderson Cancer Center

Houston, TX 77030, USA

Cancers

Zürich, 04.11.2024

Dear members of the editorial board, Dear Editors-in-Chief,

We thank the reviewer for reading our manuscript "Long-term effects on gonadal function after treatment of colorectal cancer: A systematic review and meta-analysis» and for their helpful comments and suggestions that have helped us to considerably improve the quality of our article.

We have addressed all the points raised and modified the manuscript accordingly; the revisionsare highlighted in yellow.

Please find a detailed point-by-point response below.

Comments 1:

The results of this study suggested that all treatments had the potential to damage gonadal function, albeit to varying degrees, and this is consistent with our common sense, so I think that in the Discussion section, more attention should be paid to the possible implications of the results of this study for the choice of future treatments. The clinical significance of the present study is not sufficiently persuasive in the current article.

Thank you for pointing this out. We agree with this comment. Therefore we have modified the discussion section to emphasize this point: Page 16 line 366-368 : All treatment options for colorectal cancer have the potential to damage gonadal function to varying degree, therefore we suggest considering the results of this study maybe even on a individual treatment basis for the choice of future treatments. To emphasize the clinical significance of the present study, we added this part to our discussion section page 17 line 387-389: All fertile patients with colorectal cancer should be aware of the risk of the therapy according to their fertility. This meta-analysis delivers a basis to advise all patients with colorectal cancer. 

Comments 2:

Have all included patients undergone surgery? Was the treatment analyzed in the article preoperative or postoperative? Surgery is also an important tool in the treatment of colorectal cancer, so why was the impact of surgery not considered?

Thank you for this comment. We are not sure if all patients have undergone surgery. This is not described in the sturdies. We suppose that most of them have according to international guidelines.

The treatment analyzed in the articles are preoperative and postoperative depending on the study, the stage of cancer and the treatment protocol.

Unfortunately, there were no studies that considered the impact of surgery on gonadotoxicity; they just examined the impact of radiation, chemotherapy, or both.

Comments 3:

Is it possible to conduct subgroup analyses based on age, as reproductive function, especially hormone levels, may change with age? Also, did the tumor staging have an impact on gonadal function?

Thank you for pointing this out. We agree with this comment. It is a very important point, that reproductive function changes with age. Unfortunately, it was not possible to conduct subgroup analyses based on age. The tumor staging`s impact on gonadal function was not described in the studies therefore we cannot say anything about it.

Minor issues: 

Comments 1:

The reason for the exclusion of 3514 studies in the screening process was provided in the Results section of the manuscript, and it could be clearer by adding it to Figure 1.

Thanks a lot for this comment. We just checkes Figure 1 and we already added the excluded studies as you can see on this screenshot:

Comments 2:

For the convenience of reading, it is suggested to adjust the layout format of Tables 2 and 3, and to classify the outcomes of each study according to the contents of Table 1. 

Thank you for your comment. We adjusted the layout of table 2 and 3. In the last clumn “comments” we classified the outcomes of the study according to the contents of table 1.

Comments 3:

There are some clerical errors in the manuscript, such as in Lines 195-196, “The studies were accomplished either with only men (12), only women (12), or both genders (2)”, where the sum of the total studies exceeded 22.

Thank you for pointing this out. There are studies which include only male, only women or both genders, so that we don’t have the addition of 26 studies, because some study include both men and women. That’s why the 22 studies are correct. 

Sincerely,

Dr. med. Christiane Anthon

OVA IFV, Clinic Zurich, Zurich, Switzerland

E-Mail: Christiane.wachter@gmx.de

Round 2

Reviewer 1 Report

Comments and Suggestions for Authors

Authors have taken my suggestions into consideration and revised the paper accordingly. 

Reviewer 3 Report

Comments and Suggestions for Authors

Thanks for the revision, but in my opinion, the revised manuscript is still not fully prepared for publication in Cancers. In particular, the overall novelty, clinical implication, and the control of confounding factors need further improvement.